# Effect of Vaneless Diffuser Shape on Performance of Centrifugal Compressor

**Qian Zhang, Qiuhong Huo, Lei Zhang *, Lei Song and Jianmeng Yang**

School of Energy, Power & Mechanical Engineering, North China Electric Power University, Baoding 071028, China; 51451701@ncepu.edu.cn (Q.Z.); ncepu_huoqiuhong@163.com (Q.H.); 15948694115@163.com (L.S.); 2182214098@ncepu.edu.cn (J.Y.)

* Correspondence: zhang_lei@ncepu.edu.cn; Tel.: +86-0312-752-2197

**Abstract:** The influence of four different vaneless diffuser shapes on the performance of centrifugal compressors is numerically studied in this paper. One of the studied shapes was a parallel wall diffuser. Two others had the width reduced only from hub and shroud and the rest had the width reduced from hub and shroud divided evenly. Then the numerical simulation was employed and the overall compressor aerodynamic performance was studied. The detailed velocity and pressure distribution and energy loss within the centrifugal compressor with different diffuser geometries and different operating conditions were analyzed. The results revealed that shroud pinch significantly improved the overall compressor aerodynamic performance more than any other pinch types, and the best performance can be achieved by pinched diffusers under the design condition compared with pinched diffusers under the near surge condition or choking condition. The range of energy loss, namely the static entropy area in the compressor, become reduced with the above three pinches diffusers.

**Keywords:** centrifugal compressor; vaneless diffuser; energy loss; flow field analysis

---

## 1. Introduction

Centrifugal compressors are widely used in the energy, power, aerospace, and petrochemical industries and in other fields. Improving its efficiency is still one of the difficulties and hot spots in the research of aerodynamics of turbomachinery. Optimizing the design of the components of a centrifugal compressor is an effective method that can be used to improve its performance. Xiaoqing Qiang [1] studied the influence of the volute design on the global performance of the volute, and compared ten kinds of volute design schemes by CFD (computational fluid dynamics) simulation. The final results show that, compared with the forward or symmetrical volute, the backward volute has a more reasonable and uniform velocity field and better comprehensive performance under different working conditions. Trebinjac Isabelle [2] conducted numerical simulations and experiments on a transonic centrifugal compressor and explored the unsteady physical mechanisms that affect compressor performance, such as impellers and diffusers coupling loss and the periodic changes of shock waves, and proposed an analytical model for loss correction in steady-state calculation. The instability of the rotating stall and surge under low flow conditions also greatly affects the performance of compressor [3–5]. Taher Halawa [6] carried out numerical simulations studying the rotating stall and surge conditions of centrifugal compressors with and without vanes respectively, and compared the flow characteristics of the two compressors. The results showed that the static pressure fluctuation in the vaneless diffuser was higher than that in the vaned diffuser when surge occurs, but the rate of pressure drop in the vaned diffuser is faster than that in the vaneless diffuser. Grzegorz [7] and Michele Marconcini [8] also discussed the mechanism of the rotary stall and its influence on the aerodynamic performance of the compressor through numerical simulation.

At present, the research on the centrifugal compressor impeller has been perfected, so efforts should be made to improve the performance of the centrifugal compressor from the perspective of optimizing the diffuser [9]. Vaneless diffusers [10,11] can be divided into parallel wall-types, convergent types and expansion types according to the different ratios of the diffuser outlet width to the impeller outlet width. Due to the large internal cross section of the expansion type vaneless diffuser, a large reversed pressure gradient can be generated, and secondary flows and backflows are easily generated during the flow process, so the expansion type vaneless diffuser is rarely used in practical applications.

Jaatinen-varri Ahti, Turunen-saaresti Teemu et al. [12,13] conducted experimental studies on parallel wall type vaneless diffusers with different width ratios, and the results showed that appropriate pinches of the width of vaneless diffusers at the outlet of the impeller could improve the isentropic efficiency and pressure ratio of the centrifugal compressor. The secondary flow at the shroud could be inhibited and the flow field of the diffuser could also be stabilized. A vaneless diffuser with a pinch of width led to higher performance of the centrifugal compressor stage and impeller but lower performance of the vaneless diffuser itself [14]. Yoon Yong-Sang [15] carried out a theoretical analysis of the stability of centrifugal compressors, and compared the prediction results with the experimental results of centrifugal compressors matched with three different width vaneless diffusers. Reduced width of the diffusers could make the unstable conditions appear in a smaller flow condition and reduce the number of stall cells in the rotating stall. G. Ferrara, L. Ferrari [16–18] carried out experimental studies on the stability of the final stage of a high-pressure centrifugal compressor and the performance of the compressor on the vaneless diffusers with two radius ratios, multiple width ratios, and different contraction shapes. The results confirmed that proper reduction of the width ratio can increase the stable working range of the centrifugal compressor. Although the energy loss of the system increases slightly, it has basically no effect on the overall performance of the centrifugal compressor. The effect of the change in width contraction shape at the inlet of the vaneless diffuser on the critical stall angle is within 5°. Xinqian Zheng [19] performed performance tests on axisymmetric and non-axisymmetric diffusers, and the results showed that compared with the traditional axisymmetric diffusers, the non-axisymmetric diffusers with varying design width along the circumference could extend the stable operation range of the centrifugal compressor by up to 28% at the design speed. Gao Chuang [20] explored the influence of the shape of the vaneless diffuser on the stall inception point using the wavelet neural network method, and concluded that reducing the width ratio of the vaneless diffuser and reducing the radius ratio of the wide vaneless diffuser can improve the stability of vaneless diffusers. The highest expansion stability among the width-pinch shapes of the vaneless diffuser at the exit of the three common impellers was reported with straight and oval shapes (and round was considered to be oval). The effect of the shroud pinch on the vaneless diffuser was verified by the calculation results of the vaneless diffuser 3D incompressible model developed by Chen [21], and the effect was enhanced with the increase of shrinkage rate and inlet Mach number. The study also showed that when the ratio of inlet to outlet width of the vaneless diffuser with the same radius ratio remains unchanged, for the vaneless diffuser with different contraction profiles, the more significant the inlet contraction was, the higher the stability of the centrifugal compressor would be.

In this paper, steady numerical simulation based on the RANS(Reynolds-averaged Navier-Stokes equations) equation and SST(Shear-Stress-Transport) turbulence model for a centrifugal compressor with four different vaneless diffusers and three operation conditions were studied. It could be concluded that compressor stage isentropic efficiency and pressure ratio were improved by the pinch used on vaneless diffuser width. The second important finding of the study was that implementing pinch decreases the energy loss areas within compressor. Pinch on vaneless diffuser width is relatively easy to design and manufacture, giving a novel approach to energy conservation and emissions reduction.

## 2. Numerical Method

### 2.1. Computational Model

The centrifugal compressor used in simulation is constituted by an impeller and a vaneless diffuser. Figure 1 shows the geometric model of impeller and Figure 2 shows the axial geometry of four vaneless diffusers. The major parameters of the centrifugal compressor are listed in Table 1. From published literature, the radius ratio and the reduced width of designed diffusers in this paper are all set to 1.5 and 0.2, respectively.

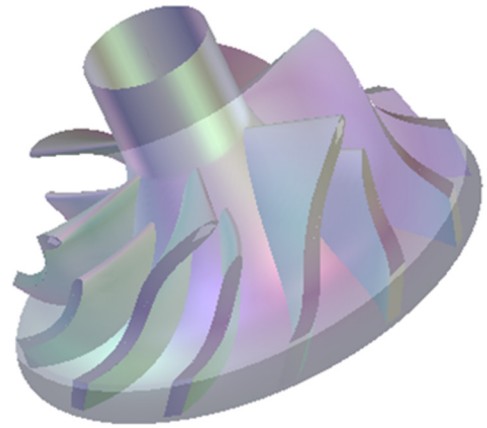

**Figure 1.** Impeller geometry model.

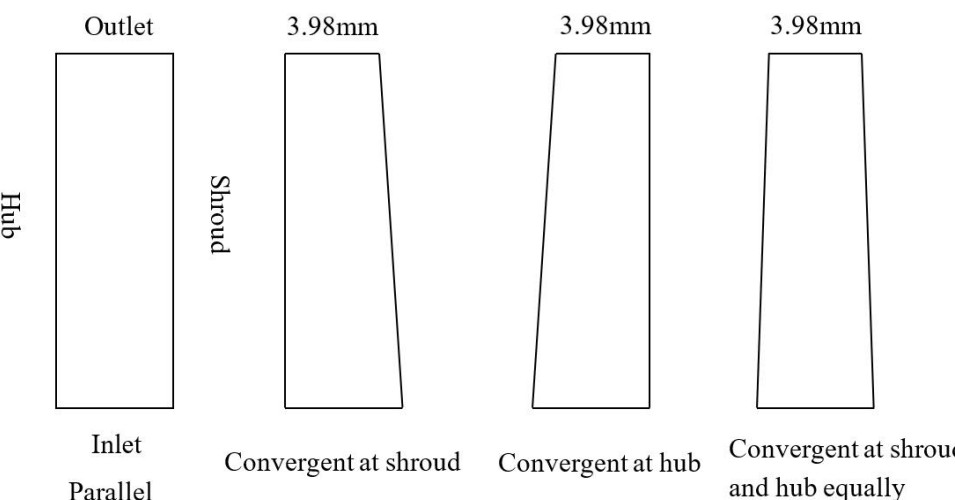

**Figure 2.** Vaneless diffuser axial geometry schematic.

The CFD numerical solver ANSYS CFX is used for solving the Reynolds-averaged Navier–Stokes equations based on the finite volume method with a second order upwind scheme for both the spatial and temporal discretization. As the basis of fluid mechanics, the N-S equation describes the general law of the velocity and pressure of compressible fluid flow. The SST model [22] is applied for turbulence closure, which is a widely used and excellent two-equation turbulence model for computational fluid dynamics. It has good computational performance when the secondary flow, flow separation and boundary layer separation occur under a large reversed pressure gradient.

**Table 1.** Centrifugal compressor major geometry parameter.

| Design Parameters | Design Value |
|---|---|
| Number of main blades | 6 |
| Number of splitter blades | 6 |
| Vaneless diffuser exit radius [mm] | 45 |
| Mass flow rate [kg/s] | 0.115 |
| Rotational speed [rpm] | 120,000 |
| Blade backsweep angle [deg] | 45 |
| Inlet tip radius [mm] | 21.4 |
| Impeller exit tip radius [mm] | 30 |
| Relative tip clearance | 0.02 |
| Relative outlet width of impeller | 0.166 |
| Tip diameter reynolds number | $1.43148725 \times 10^6$ |

### 2.2. Mesh Division and Boundary Condition Setting

In this paper, the wall surface is set to adiabatic and non-slip boundary conditions and frozen rotor interface is selected to connect the static interface between the impeller and the diffuser. The inlet surface is given a total temperature of 288.15 K and a total pressure of 101,325 Pa. Pressure inlet and mass flow outlet conditions are set at the inlet and outlet surface of the centrifugal compressor, respectively. Three types of mass flow are set in order to investigate the influence in detail, which are 60%, 100%, and 112.5% of the design flow. The rotational speed of the impeller is the design speed (120,000 rpm). The ideal gas medium is selected for the internal medium to ignore the changes in gas density and the effect of heat exchange. The momentum equation, continuity equation and energy equation were solved by coupling solver. The difference scheme of high resolution controlled the accuracy of the solution, and the convergence criterion was set to the residual value less than $10^{-4}$.

The impeller computational domain was discretized automatically using structured mesh by means of a component of CFX named Turbogrid and the diffuser computational domain was divided by software Gambit. To ensure that the value of Y+ near the diffusers wall was around 1, the boundary layer grids of vaneless diffusers were set to fifteen layers, and grow rate was set to 1.2. Then mesh independence verification was proposed in order to make sure the result of simulation correctly. By changing the minimum and maximum mesh number of compressor, three mesh sets were implemented with a total number of 1.5 million up to 3.5 million cells. The effects of the compressor grid on the calculation performance are shown in Table 2. It can be seen that the pressure ratio difference of fine grid level and medium grid level is smaller than the difference between medium grid level and coarse grid level from Table 2. For the limitation of calculation source, the medium grid level concluded 2.33 million cells was selected in this article.

**Table 2.** Effects of the compressor grid on the calculation performance.

| Grid Level | Total Cells | Pressure Ratio |
|---|---|---|
| Coarse | 1.5 Million | 2.138 |
| Medium | 2.33 Million | 2.251 |
| Fine | 3.5 Million | 2.254 |

### 2.3. Validation of the Numerical Model

For the limitation of the experimental conditions, the numerical model in this paper was validated by comparing the result of simulation with the experimental data of Krain's impeller from the opening literature [23] in the authors' previous paper [24]. Krain's impeller is a transonic compressor impeller designed by Krain et al. [25]. The comparison of pressure ratio between experimental and numerical simulation is presented in Figure 3. From Figure 3, the result of simulation is generally less than the experimental data, and the deviation between two curves caused by the contradiction between the

number of grids and the calculation cost or the description of the real flow by the numerical algorithm is within 6%. So it can be concluded that the numerical scheme in this paper can predict the compressor performance accurately.

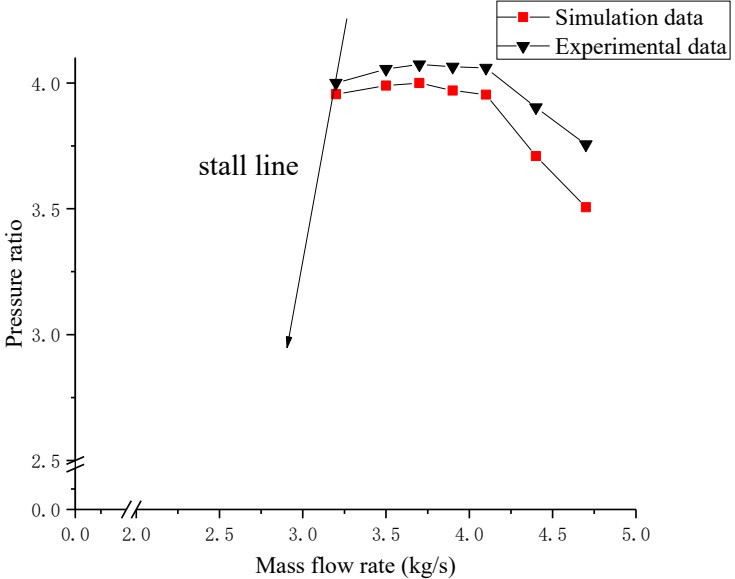

**Figure 3.** Comparison of calculated and experimental total pressure ratios.

## 3. Result and Discussion

### 3.1. Aerodynamic Performance Analysis

The ratio of actual flow conditions to design flow conditions is defined as relative mass flow $\varphi/\varphi_D$. It can be known from Table 3 that under the design conditions and 0.6 relative mass flow conditions, the whole stage pressure ratio can be increased through reducing diffuser width and the best effect on the pressure ratio improvement can be achieved by convergence of the vaneless diffuser width from the shroud side. However, the convergence of the diffuser width under the 1.125 relative mass flow condition causes the pressure ratio of the centrifugal compressor to decrease. This phenomenon is similar to the experimental results in literature [9], indicating that the positive effect cannot be achieved in making use of the convergence of the vaneless diffuser width under all conditions.

**Table 3.** Total-to-total pressure ratio of centrifugal compressor.

| Relative Mass Flow $\varphi/\varphi_D$ | Vaneless Diffuser Shape | Press Ratio $p_4/p_1$ |
|---|---|---|
| 1.125 | Parallel | 1.914 |
| 1.125 | Convergence from shroud | 1.905 |
| 1.125 | Convergence from hub | 1.903 |
| 1.125 | Convergence from shroud and hub equally | 1.904 |
| 1.0 | Parallel | 1.931 |
| 1.0 | Convergence from shroud | 1.970 |
| 1.0 | Convergence from hub | 1.964 |
| 1.0 | Convergence from shroud and hub equally | 1.967 |
| 0.6 | Parallel | 2.309 |
| 0.6 | Convergence from shroud | 2.342 |
| 0.6 | Convergence from hub | 2.339 |
| 0.6 | Convergence from shroud and hub equally | 2.341 |

In order to investigate which part of the centrifugal compressor is affected by the convergent diffuser width, the inlet and outlet pressure ratios of the impeller and the diffuser are extracted, as

shown in Table 4. The convergence of the vaneless diffuser width can increase the impeller outlet pressure and has little effect on the change of the vaneless diffuser pressure ratio. Furthermore, it can be found that when the relative mass flow is 1.125, the impeller pressure ratio with convergent diffuser increases slightly but the pressure ratio of the convergent diffuser decreases compared with parallel wall diffuser case. This may be the reason for the convergence of the diffuser width but the increase of the centrifugal compressor pressure ratio in the near choking condition.

**Table 4.** Centrifugal compressor impeller and diffuser pressure ratio.

| Relative Mass Flow $\varphi/\varphi_D$ | Vaneless Diffuser Shape | Press Ratio $p_2/p_1$ | Press Ratio $p_4/p_3$ |
|---|---|---|---|
| 1.125 | Parallel | 1.865 | 1.026 |
| 1.125 | Convergence from shroud | 1.872 | 1.018 |
| 1.125 | Convergence from hub | 1.876 | 1.014 |
| 1.125 | Convergence from shroud and hub equally | 1.874 | 1.016 |
| 1.0 | Parallel | 1.878 | 1.028 |
| 1.0 | Convergence from shroud | 1.915 | 1.029 |
| 1.0 | Convergence from hub | 1.913 | 1.026 |
| 1.0 | Convergence from shroud and hub equally | 1.915 | 1.027 |
| 0.6 | Parallel | 2.252 | 1.025 |
| 0.6 | Convergence from shroud | 2.288 | 1.023 |
| 0.6 | Convergence from hub | 2.290 | 1.022 |
| 0.6 | Convergence from shroud and hub equally | 2.289 | 1.023 |

Table 5 shows the isentropic efficiency of centrifugal compressors with different diffuser shapes under multiple operating conditions. Isentropic efficiency is the ratio of the isentropic power of the compressor to the actual power required by the compressor. It can be concluded from Table 5 that all three convergent shape vaneless diffusers improve the isentropic efficiency of centrifugal compressors. The maximum isentropic efficiency can be reached by converging the diffuser wall from the hub side at 1.125 and 1.0 relative mass flow conditions and from shroud side at 0.6 relative mass flow condition. The change of isentropic efficiency in the shape of the vaneless diffuser is also consistent with the experimental phenomenon in literature [9].

**Table 5.** Centrifugal compressor isentropic efficiency.

| Relative Mass Flow $\varphi/\varphi_D$ | Vaneless Diffuser Shape | Isentropic Efficiency [%] |
|---|---|---|
| 1.125 | Parallel | 81.04 |
| 1.125 | Convergence from shroud | 81.28 |
| 1.125 | Convergence from hub | 81.36 |
| 1.125 | Convergence from shroud and hub equally | 81.35 |
| 1.0 | Parallel | 82.12 |
| 1.0 | Convergence from shroud | 82.75 |
| 1.0 | Convergence from hub | 82.81 |
| 1.0 | Convergence from shroud and hub equally | 82.78 |
| 0.6 | Parallel | 72.24 |
| 0.6 | Convergence from shroud | 72.62 |
| 0.6 | Convergence from hub | 72.44 |
| 0.6 | Convergence from shroud and hub equally | 72.53 |

*3.2. Flow Field Analysis*

3.2.1. Design Condition Meridian Velocity Streamline

Figure 4 shows the meridional velocity streamlines for the four diffusers design conditions. The position and size of the vortexes inside the vaneless diffusers get greatly affected by the convergence of the width of the vaneless diffusers. The tail vortex area from the radial and axial directions within compressors is suppressed in all three pinch shapes of diffusers. Compared with the parallel wall diffuser, the flow cross-section of the fluid in the diffusers with pinch width is reduced,

which decreases the pressure gradient inside the diffusers and makes the flow within compressors relatively uniform. Not only the backflow phenomenon in the diffusers are improved, but also the backflow vortexes at the impeller outlet are also suppressed, which make the compressor internal pressure distribution more uniform. Analysis of Figure 4 shows that the convergence of the diffuser width from the shroud side has the best effect on suppressing the return vortex phenomenon.

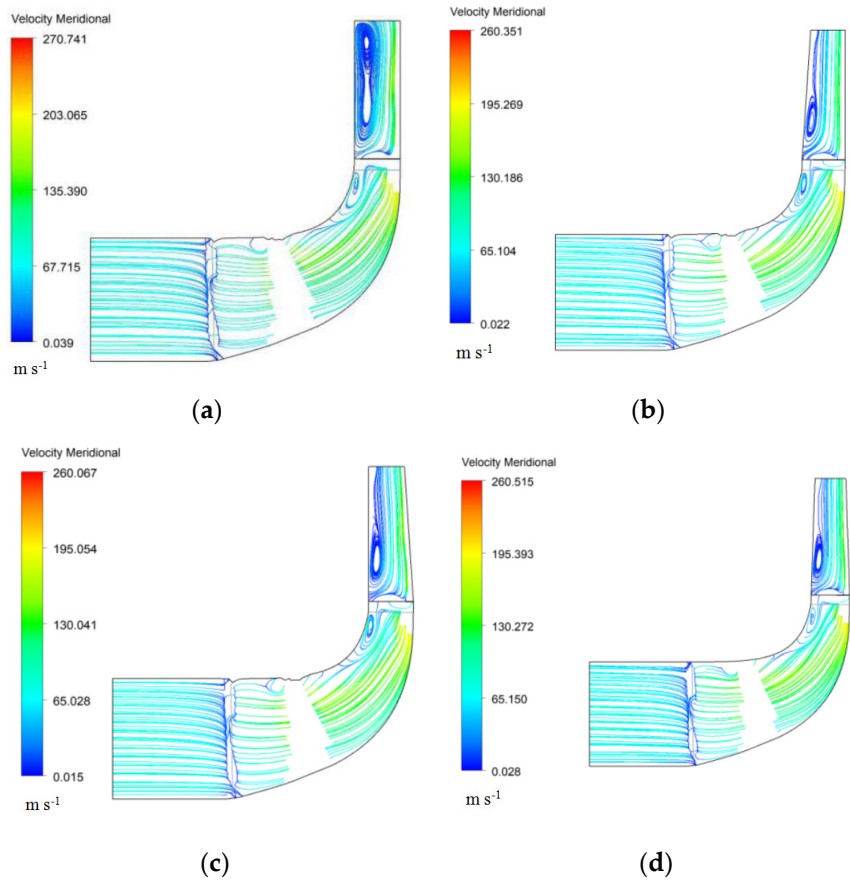

**Figure 4.** Radial velocity streamline distribution of the entire meridian at the design mass flow: (**a**) parallel; (**b**) convergence from shroud; (**c**) convergence from hub; (**d**) equal convergence from shroud and hub.

### 3.2.2. Meridian Plane Radial Velocity Distribution

Through the analysis of Figure 4, the shape of the vaneless diffuser has a great influence on the flow stability under the design conditions of the centrifugal compressor, especially the internal flow field of the vaneless diffuser. In order to analyze the internal flow characteristics of the diffuser in detail, the method of analyzing the radial velocity distribution in the meridian plane is adopted.

Figure 5 shows the radial velocity distribution on the meridional surface of the vaneless diffuser at 1.125 relative mass flow operation condition. Both the minimum and maximum radial velocity of the meridional surface are observed in the parallel wall vaneless diffuser. The low-speed backflow area appears on the shroud side of the parallel wall diffuser is suppressed after converging the width, and the backflow phenomenon is improved. The high-speed core flow appearing on the hub side of the parallel wall diffuser is also affected along with the convergence of the width. The smaller the high-speed region indicates that the internal speed of the diffuser changes to pressure and the efficiency of the diffuser becomes higher. The backflow area at the position where the impellers are connected to the vaneless diffusers is unaffected as the width convergence. From Figure 5, it can be concluded that converging the wall of the vaneless diffuser from the shroud side is the best way to improve the low-speed return flow area near shroud.

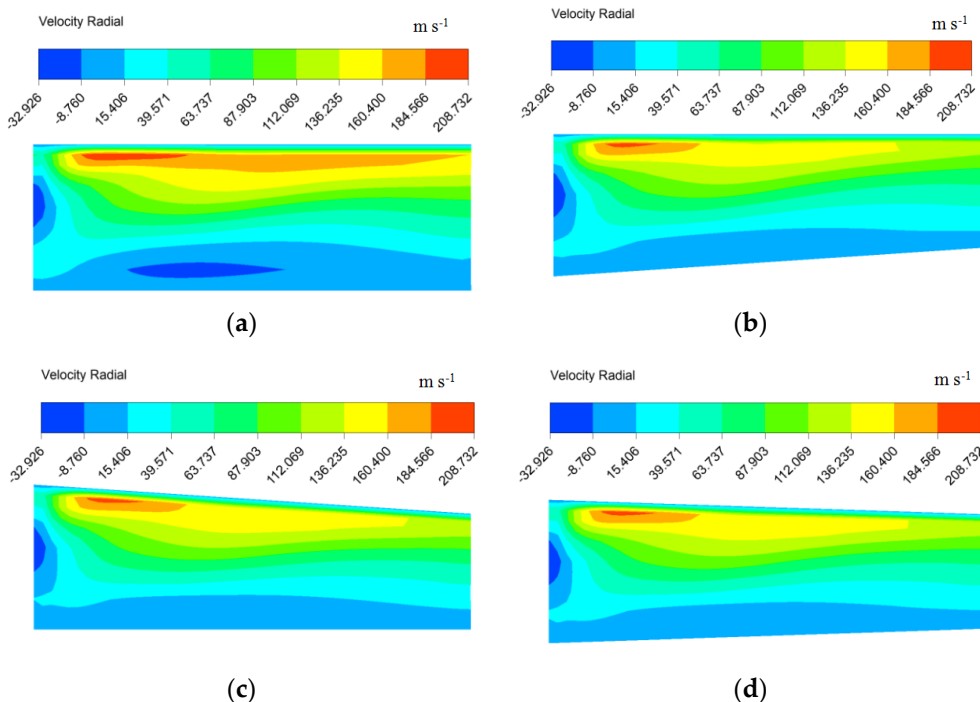

**Figure 5.** Radial velocity distribution on the meridional surface at 1.125 relative mass flow: (**a**) parallel; (**b**) convergence from shroud; (**c**) convergence from hub; (**d**) equal convergence from shroud and hub.

Figure 6 shows the radial velocity distribution of the meridional surface of the four vaneless diffusers under design conditions. Compared with the relative mass flow of 1.125, the low-speed return flow area on the shroud side of the parallel wall diffuser extends from the shroud side entrance to the exit and another high speed zone is located near the shroud side exit. The flow field in the three types of diffuser is changed similarly to that when the relative mass flow is 1.125. The backflow area on the shroud side basically disappears and the high-speed flow area on the hub side becomes smaller.

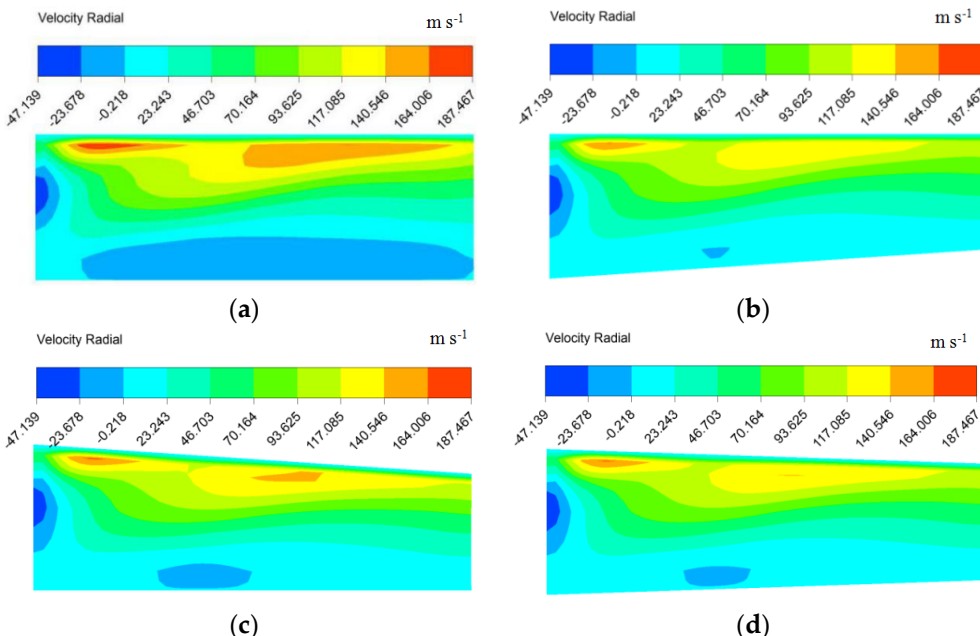

**Figure 6.** Radial velocity distribution on the meridional surface at 1 relative mass flow: (**a**) parallel; (**b**) convergence from shroud; (**c**) convergence from hub; (**d**) equal convergence from shroud and hub.

Figure 7 shows the radial velocity distribution of the meridional plane in four vaneless diffusers at 0.6 relative mass flow operation condition. As the relative mass flow decreases, the instability of the flow intensifies, and the value of the backflow area at the connection between the vaneless diffuser and the impeller is lower, and the backflow area increases along the span direction. What is different from the relative mass flow of 1.125 and 1.0 is that the convergence of the width can significantly suppress the backflow phenomenon namely reduce the backflow area at the connection between the impeller and the vaneless diffuser.

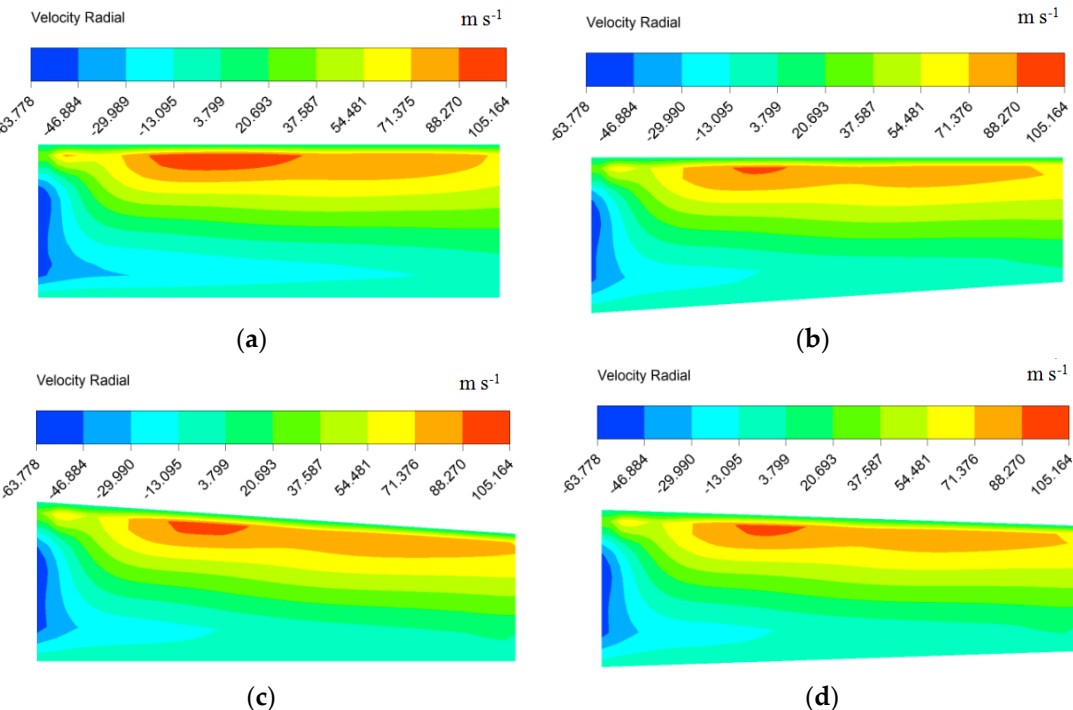

**Figure 7.** Radial velocity distribution on the meridional surface at 0.6 relative mass flow: (**a**) parallel; (**b**) convergence from shroud; (**c**) convergence from hub; (**d**) equal convergence from shroud and hub.

The backflow area near the shroud side is extended almost from the inlet to the near exit of the vaneless diffusers along the length direction. Various shapes of diffuser convergence cannot completely eliminate backflow area on the shroud side, but can suppress the expansion of the backflow area from the length of the vaneless diffuser.

3.2.3. Span Cross-Section Radial Velocity

The radial velocity distribution was analyzed by taking 70% of the span cross-section locations with a large return flow zone near the shroud side to display the circumferential flow characteristics of the vaneless diffusers.

Figures 8–10 show the radial velocity distribution inside vaneless diffusers at 70% span height under three flow conditions. As can be seen from Figure 8 that when the relative mass flow is 1.125, the backflow phenomenon at the exit of the impeller is aggravated by the convergence of the vaneless diffuser width, and the value of the high-speed area of the circumferential flow is increased, which may be the reason why the pinch of diffuser width will lead to the decline of the compressor pressure ratio under the condition of near choking. The most negative impact investigated from Figure 8 is the case that converges the width of the diffuser from the hub side. What causes this is the flow high-speed area that near the hub side. Convergence of the diffuser width from the hub side under the condition of large mass flow tends to increase the radial speed value of the overall high-speed area and reduce the conversion process of velocity to pressure in the case of large mass flow.

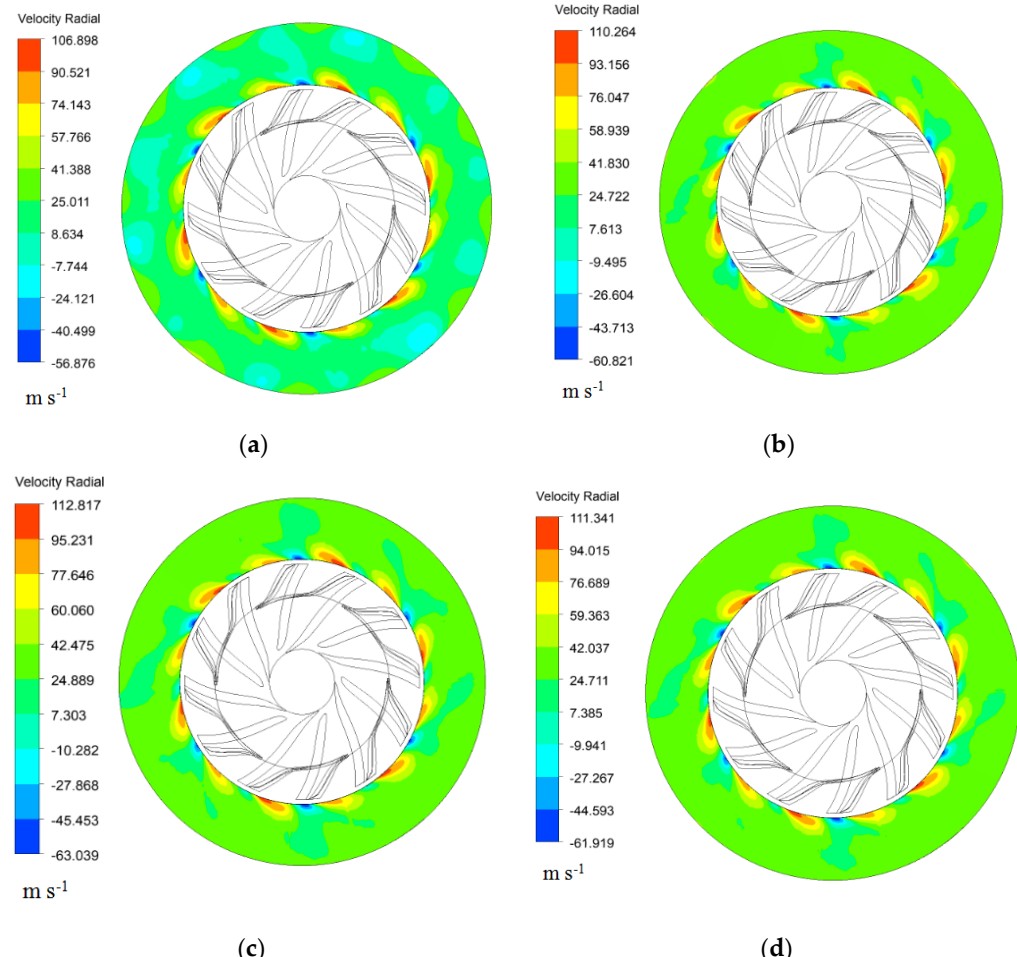

**Figure 8.** Seventy percent span height radial velocity distribution at 1.125 relative mass flow: (**a**) parallel; (**b**) convergence from shroud; (**c**) convergence from hub; (**d**) equal convergence from shroud and hub.

The three convergent forms of vaneless diffuser can improve the backflow area at the outlet of the impeller and decrease the value of the high-speed flow area under the design flow condition, as shown in Figure 9. Among the three shapes, the width of the diffuser can be pinched from the shroud side to the best effect. Compared with the relative mass flow of 1.125, the width of the three convergent diffusers can effectively reduce the backflow area on this section, and the vaneless diffuser that converged from the shroud of the wall under this flow condition makes the backflow area shrink most significantly.

For the 0.6 relative mass flow operation condition, the jet wake structure located at the impeller outlet in the parallel wall diffuser is suppressed by the downstream backflow area, and distortion acted on jet wake structure occurs follow on, as shown in Figure 10. After converging the diffuser width, the backflow zone disappears and the jet wake structure recovers, indicating that the convergence of the width improves the flow characteristics at the diffuser inlet, and besides these the three convergence forms have little difference in effect on the backflow area.

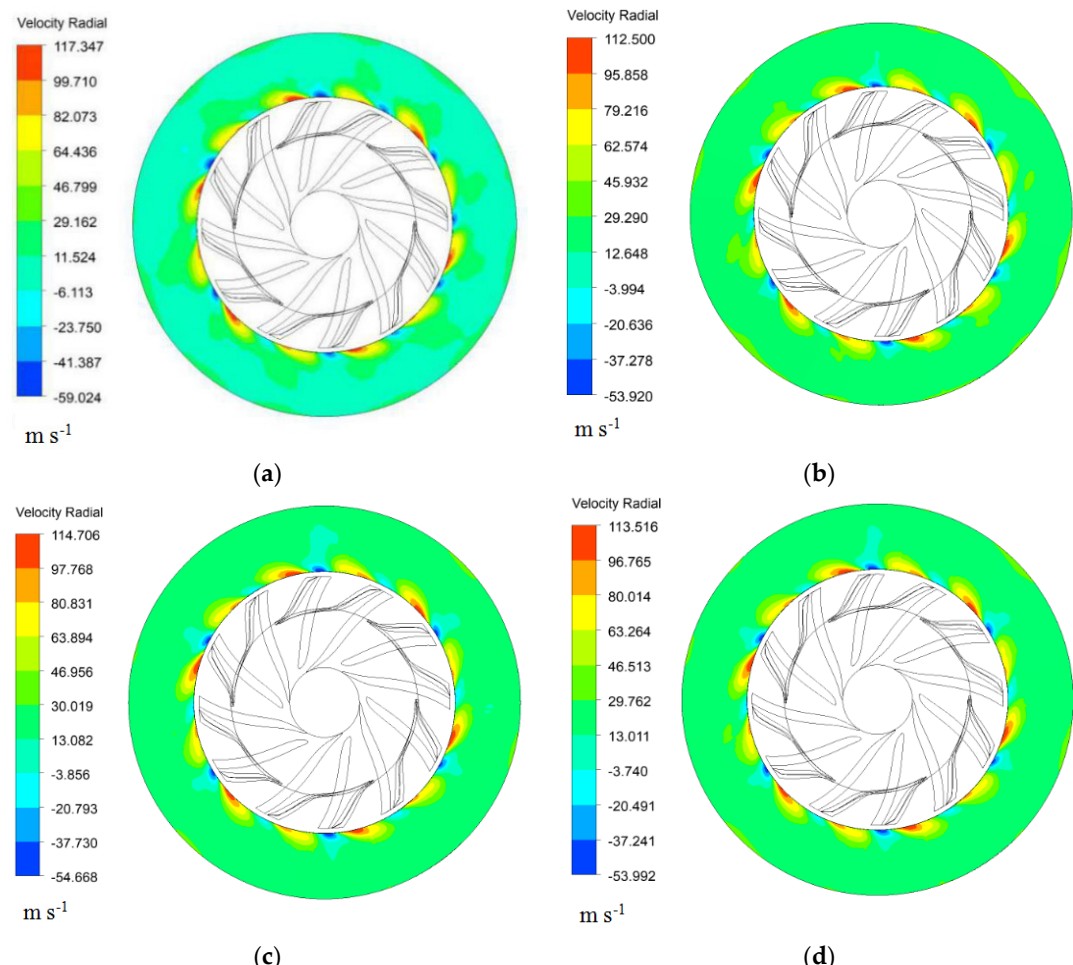

**Figure 9.** Seventy percent span height radial velocity distribution at 1 relative mass flow: (**a**) parallel; (**b**) convergence from shroud; (**c**) convergence from hub; (**d**) equal convergence from shroud and hub.

Through combining the radial velocity profile of the diffuser meridian and the 70% span height cross section contours, it can be seen that the convergence of the vaneless diffuser width can significantly improve the backflow phenomenon on the shroud side and reduce the high speed flow area on the hub side when the relative mass flow is 1.125 and 1. At 0.6 flow coefficient, there is no significant difference in the effect of the three convergent forms of the vaneless diffuser on the backflow area on the meridian near the shroud side, but the size of the this area where the impeller is connected to the diffuser can be reduced. Therefore, the efficiency improved by the convergence of the diffuser width is summarized as suppressing the internal backflow area of centrifugal compressor.

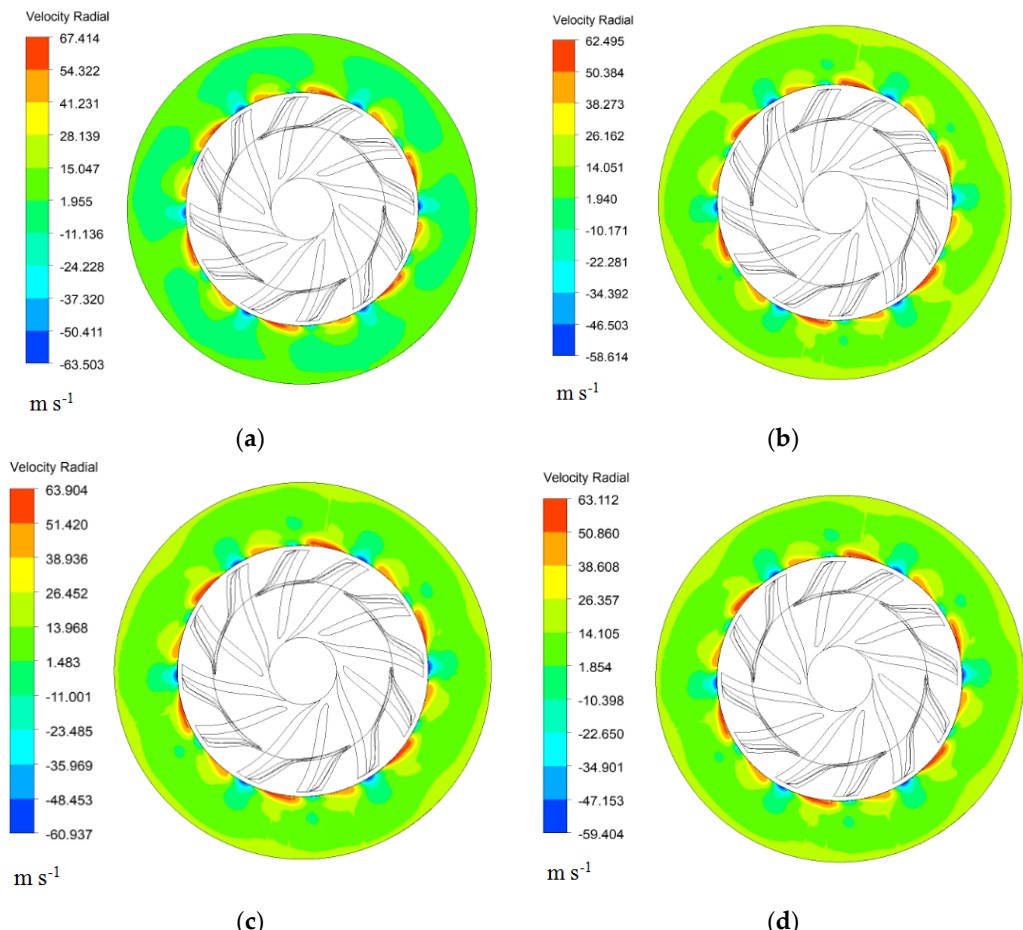

**Figure 10.** Seventy percent span height radial velocity distribution at 0.6 relative mass flow: (**a**) parallel; (**b**) convergence from shroud; (**c**) convergence from hub; (**d**) equal convergence from shroud and hub.

### 3.2.4. Pressure Distribution Law of Diffusers in Length Direction

Near the entrance of the vaneless diffusers in Figure 11, the difference between the static pressure values of the three convergent vaneless diffusers and the parallel wall vaneless diffuser has no significant difference. And in Figure 11, the definition of symbols below picture can be find in Appendix A. The static pressure value always shows a convergent type with the extension of the diffuser length. The state pressure of diffusers of reduced width is higher than that of the parallel wall diffuser and the static pressure difference is gradually increased. The variation of static pressure of the three convergent vaneless diffusers is formed a stable law after 15% of the vaneless diffuser length: the highest static pressure can be realized by converging diffuser wall from the shroud side, the lowest one can be realized by converging from the hub side.

As for the total pressure distribution, the parallel wall diffuser and the three forms of convergent vaneless diffusers have the same total pressure distribution trend along the length of the diffusers, and the total pressure distribution of the three forms of convergent diffusers are always higher than the parallel wall diffuser. Due to the complexity of the flow field in the vaneless diffuser, such as the jet-wake structure at the impeller outlet, the clearance leakage vortex, and the boundary layer separation in the vaneless diffuser, the total pressure of the vaneless diffuser gradually decreased in the radial direction.

The static pressure rise effect of the three convergent forms of diffusers in Figure 12 is more obvious than that when the relative mass flow is 1.125 in Figure 11. Obviously, the effect of diffuser convergence on static pressure and total pressure distribution tends to be consistent.

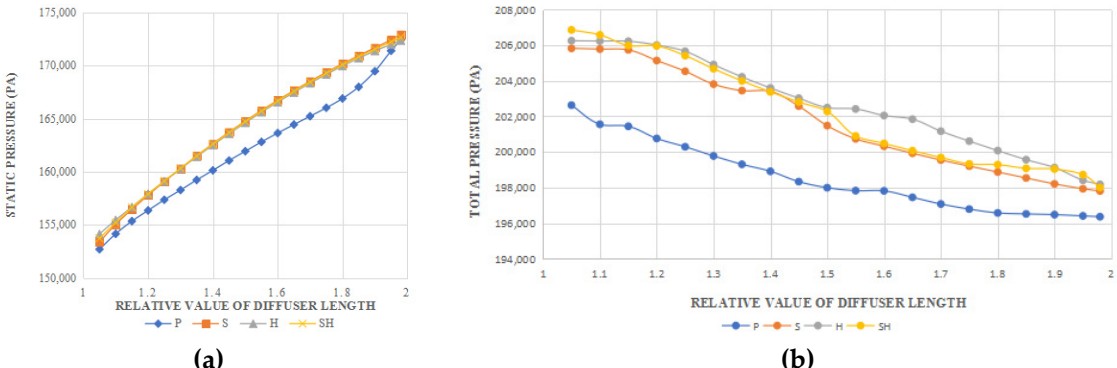

**Figure 11.** Mean static pressure and total pressure distribution along the diffuser length direction at 1.125 relative mass flow: (**a**) static pressure distribution; (**b**) total pressure distribution.

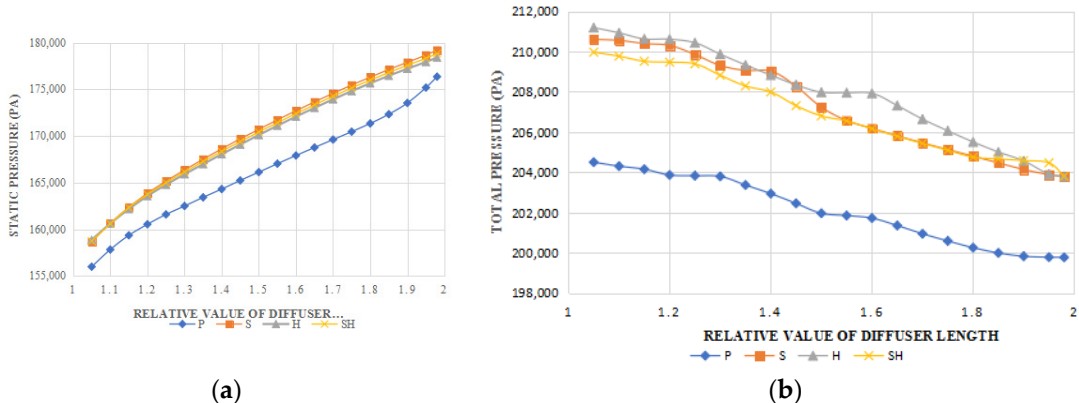

**Figure 12.** Mean static pressure and total pressure distribution along the diffuser length direction at 1 relative mass flow: (**a**) static pressure distribution; (**b**) total pressure distribution.

Figure 13 shows the mean distribution of static pressure and total pressure along the length direction of vaneless diffusers at 0.6 flow coefficient. The static pressure distribution under these conditions maintains the same trend for the four vaneless diffusers, and the static pressure rise is not as significant as the design conditions. It can be seen that the total pressure value is obviously larger than the other two working conditions. The performance of static pressure and total pressure in these three convergent vaneless diffusers is still consistent with the previous two conditions.

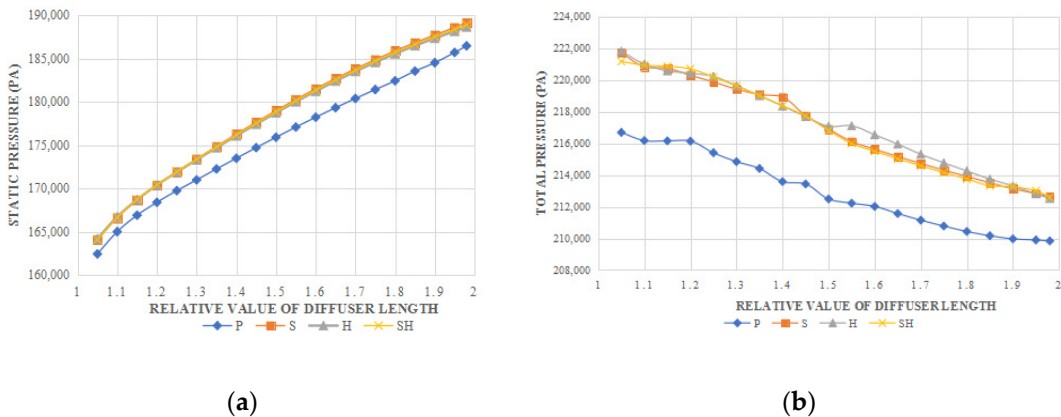

**Figure 13.** Mean static pressure and total pressure distribution along the diffuser length direction at 0.6 relative mass flow: (**a**) static pressure distribution; (**b**) total pressure distribution.

The conclusion for the distribution of total pressure and static pressure along the length direction of the vaneless diffuser in three operating conditions can be summarized as follows: the best effect is reached through the pinch from the shroud side, and the advantage of this shape is more obvious under the design operating condition. The improvement of the static pressure of the vaneless diffuser by the convergence of the width is mainly reflected in the second half of the length of the diffuser. The difference in the total pressure is mainly reflected in the pressure value along the vaneless diffusers, especially under 0.6 relative mass flow condition.

*3.3. Energy Loss*

With the development of CFD technology, it is possible to express the energy loss mechanism inside a centrifugal compressor through the change of entropy. The smaller the increase in entropy under the same working conditions, the smaller the degree of energy loss, and the higher the efficiency of the system. In this paper, the degree of energy loss in the system is measured by the change of the static entropy area of the meridional plane under three different conditions of four vaneless diffusers.

The meridional surface static entropy distribution under three operation conditions is demonstrated in Figures 14–16. The same scale is used in these contours. For the static entropy distribution on the meridional surface of the diffusers at 1.125 relative mass flow in Figure 14, there is an increased static entropy area at the exit of the parallel wall vaneless diffuser compared with the three diffusers with convergent width, and the area where the entropy value near the hub side develops from the lowest range to a higher range is wider. The energy loss in the span direction is also greater near the shroud side. Inside the impeller, there is no obvious difference in the distribution of the static entropy of the four diffusers. Under this condition, the effect of shape of vaneless diffuser on the internal energy loss of the impeller is relatively small, which is similar to the above conclusion. There is no obvious change in the form of the diffuser. At the connection between the vaneless diffuser and the impeller, the increase of entropy area with the parallel wall vaneless diffuser is more obvious. Moreover, the effect of the three convergent forms of vaneless diffuser on the reduction of energy loss in this operating condition is not much different.

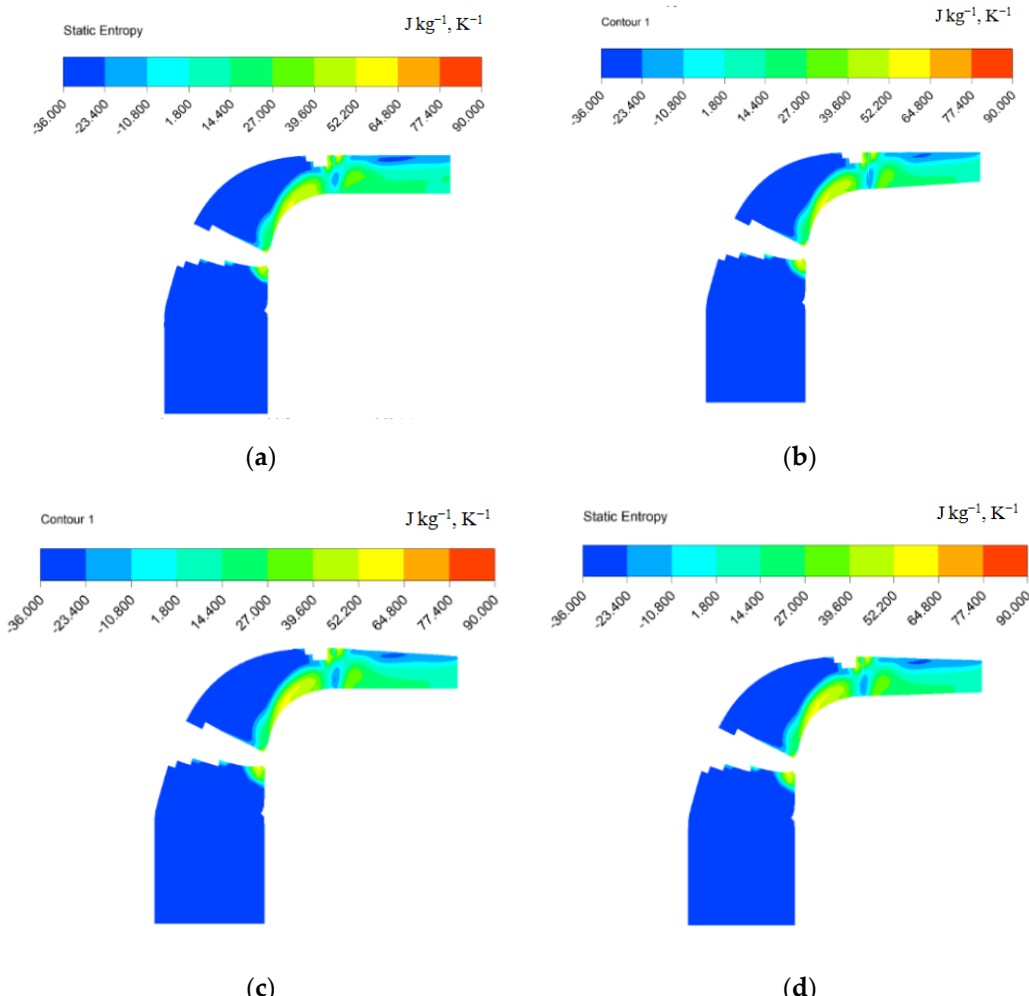

**Figure 14.** Meridional surface entropy distribution at 1.125 relative mass flow (**a**) parallel; (**b**) convergence from shroud; (**c**) convergence from hub; (**d**) equal convergence from shroud and hub.

The difference in the entropy distribution between the parallel wall diffuser and the convergent diffusers is larger than that when the relative mass flow is 1.125 under the design flow conditions, as shown in Figure 15. A significantly larger range of entropy area and amplitude of entropy rise near the impeller exit of the parallel wall diffuser shroud side. The energy loss of the three convergent vaneless diffusers at this location is relatively small, and there is no significant difference in effect.

When the relative mass flow is 0.6, the meridional entropy distribution of the centrifugal compressor is shown in Figure 16. The internal entropy distribution of the centrifugal compressor impeller is significantly different from the previous two flow conditions, and the trend of entropy rise is more significant. A larger entropy increases range near the shroud side of the parallel section of the impeller flow channel of the parallel wall vaneless diffuser more than the other three vaneless diffuser types, which shows that the width convergence can reduce the energy loss here. The internal entropy value of the parallel wall diffuser has a larger development range from low to high in the span direction and length direction, and the range of entropy rise in hub side has been extended to the exit of diffuser.

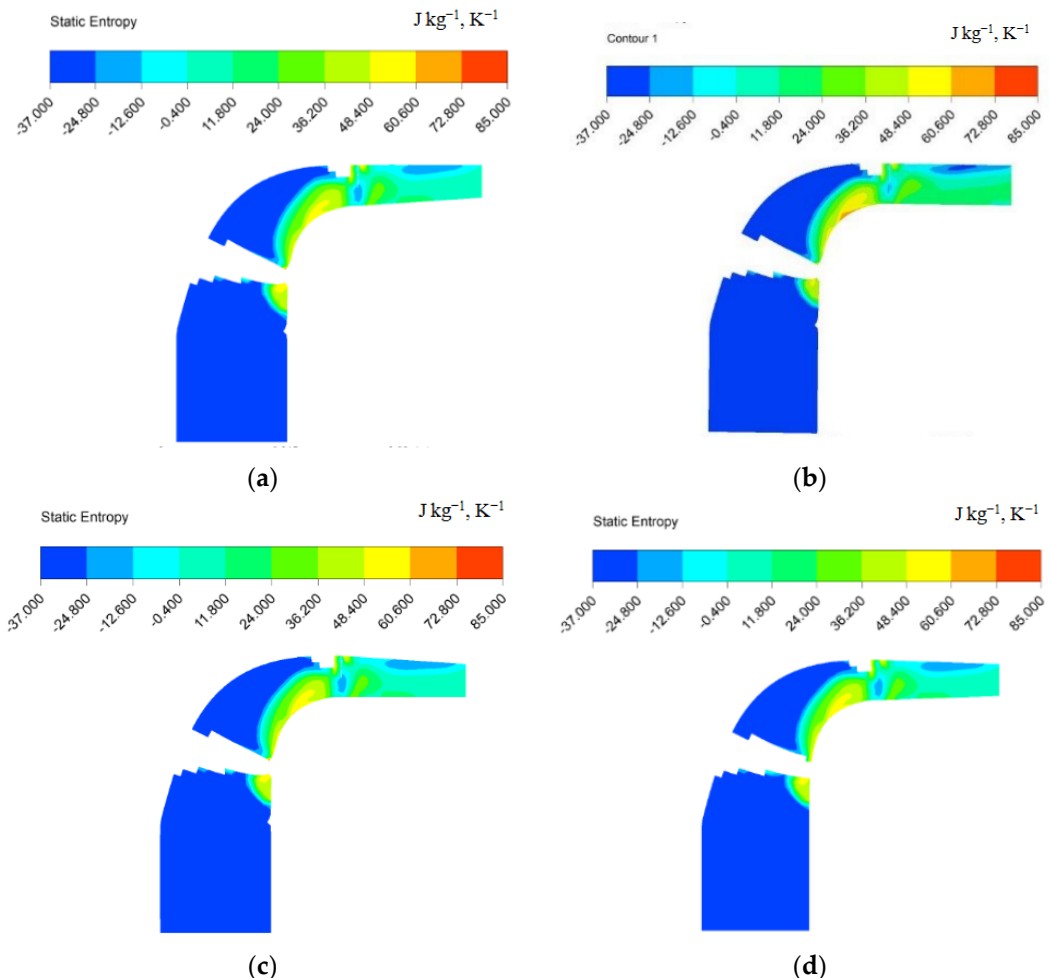

**Figure 15.** Meridional surface entropy distribution at 1 relative mass flow (**a**) parallel; (**b**) convergence from shroud; (**c**) convergence from hub; (**d**) equal convergence from shroud and hub.

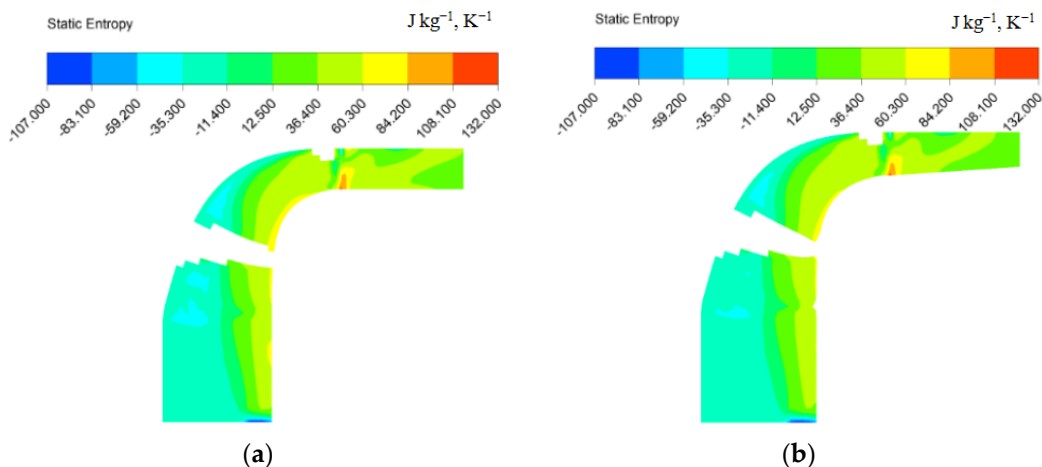

**Figure 16.** *Cont.*

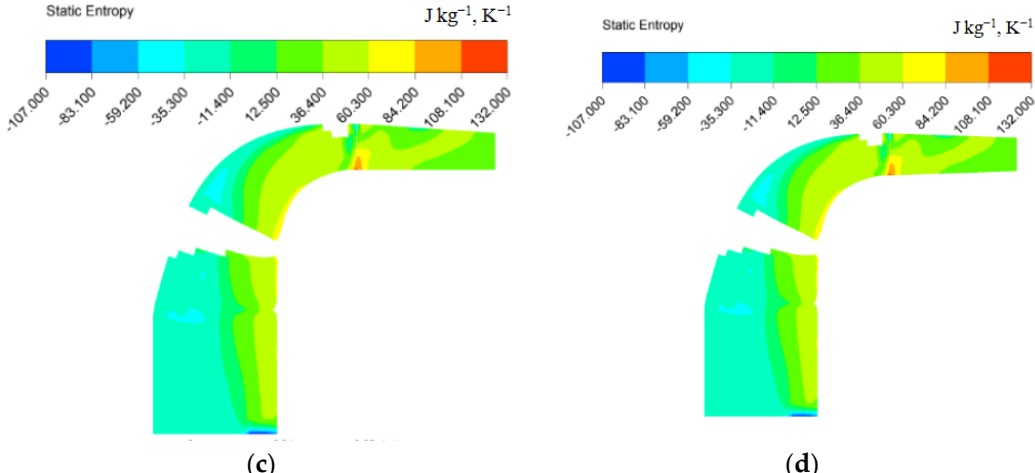

**Figure 16.** Meridional surface static entropy distribution at 0.6 relative mass flow: (**a**) parallel; (**b**) convergence from shroud; (**c**) convergence from hub; (**d**) equal convergence from shroud and hub.

## 4. Conclusions

The centrifugal compressors with four different shape vaneless diffusers were numerical simulated in this paper. The shapes of the diffuser were modified by proper pinch on the shroud side and hub side of the vaneless diffuser wall. Based on the CFD simulation results, the aerodynamic characteristics and internal flow characteristics of the centrifugal compressor were studied. The conclusions were summarized as follows:

(1) Three convergent vaneless diffuser shapes mentioned in this article can improve the overall stage pressure ratio and isentropic efficiency of centrifugal compressors under design condition and small flow rate condition. The pinch on the diffuser shroud side can achieve the optimal expansion effect of pressure ratio and isentropic efficiency under small flow rate condition and design condition, and the expansion effect is concentrated on the impeller, while the pinch on the diffuser in the near choking condition produces negative effect for the pressure ratio of compressor. As for the isentropic efficiency, convergence on the shroud side is the best way to improve efficiency under small flow rate condition, while convergence on the hub side is the best way to improve efficiency under design condition and near choking condition.

(2) By analyzing the internal flow field of the vaneless diffusers, the convergence of the diffuser width can optimize and improve the internal backflow phenomenon, thereby increasing the efficiency of compressors. Although the effect of the diffuser modification on the pressure ratio of the diffuser is not obvious, the corresponding change can make the pressure change of diffusers along the length direction more uniform and reduce the internal reverse pressure gradient.

(3) The internal energy loss is changed significantly with the change of shapes of the vaneless diffuser. Therefore, the change of the static entropy distribution range in the contour is observed to judge the variation of the energy loss. The effect of convergent diffuser width is not obvious in the near choking condition, while the convergent diffuser width under the design condition and small flow rate condition can effectively reduce the area of entropy rise within the compressor, so as to reduce energy loss and improve efficiency. There is no significant difference of the three convergent forms mentioned above on the influence of energy loss.

**Author Contributions:** Conceptualization, Q.Z.; methodology, Q.Z., L.Z. and J.Y.; writing—original draft, Q.H. and Q.Z.; writing—review and editing, L.Z.; software, L.S. All authors have read and agreed to the published version of the manuscript.

**Funding:** This research was supported by National Natural Science Foundation of China (Grant No.11602085), the Fundamental Research Funds for the Central Universities of China (Grant No. 2018MS107), and the Natural Science Foundation of Hebei Province, China (Grant No. E2016502098).

**Conflicts of Interest:** The authors declare no conflict of interest.

## Appendix A  Explanation of Main Symbols in Article

| | |
|---|---|
| Φ | Flow coefficient (Impeller exit actual mass flow/design mass flow) |
| 1 | Impeller inlet |
| 2 | Impeller outlet |
| 3 | Vaneless diffuser inlet |
| 4 | Vaneless diffuser outlet |
| P | Parallel wall diffuser |
| S | Convergence from shroud |
| H | Convergence from hub |
| SH | Equal convergence from shroud and hub |

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
