# Peer review of "Effect of Vaneless Diffuser Shape on Performance of Centrifugal Compressor"

_applsci, doi:10.3390/app10061936_

Round 1

Reviewer 1 Report

After a careful revision of the paper by the authors, I recommend acceptance for journal publication.

A list of detailed comments is given below. Among others, the comments are meant to increase the clarity of the paper.

a)Some used impeller parameters are not carefully defined. I miss

- relative outlet width of impeller b2/r2

- Reynolds number, e.g.   Re = U2 D2

- definition of isentropic efficiency .

b) The definition of flow coefficient in paper is somewhat confusing. The specialists in internal aerodynamics use generally the definition of flow coefficient φ as a ratio of flow velocity and peripheral one. Reviewer recommends to use the notations: relative flow coefficient φ/ φD or relative mass flow m/mD , m – mass flow rate (kg/s).

c) flow stability in vaneless diffuser is often investigated with the use of the value of absolute flow angle at diffuser inlet. In literature there are some critical flow angle values for flow instability origin : 76o to 85o to the radial. Could authors present the flow angle values valid for all studied compressor working conditions.

d) The dependencies of total pressure on normalized distance from the diffuser inlet are drawn  in Figs No. 11 to 13 for 3 compressor working points. It can be observed an increase of total pressure near inlet and outlet parts of vaneless diffuser. Could authors explain this phenomenon?

e) Some information about flow solution convergency is necessary. Which criterion was applied?

Authors present the values of isentropic efficiency with accuracy of 2 digits behind decimal point. Is this in coincidence with CFD computation accuracy?

Reviewer 2 Report

The paper presents the numerical of centrifugal compressors with four vaneless diffusers of different geometries. The aerodynamic characteristics and the internal flow characteristics (velocity, pressure distribution, energy losses) of the centrifugal compressor are studied based on the CFD simulation results.

The paper fits the journal topics, being organized in a logical manner. The state of art covers the latest results in the field. The authors’ own results are included in the state of art. The results of the numerical simulations are discussed and conclusions regarding the influence of the diffusers geometry on the characteristics of the centrifugal compressors are formulated.

There are some grammar errors and typos in the text that should be corrected (…of a centrifugal compressor stage is numerical studied …; Optimizing the design and test of components of centrifugal compressor is an effective method can be used to improve the performance; … isentropic efficiency and pressure ratio was improved by …; .. is relatively easy to design and manufacturing, giving …; For fully display the circumferential flow characteristics of the vaneless diffusers, the radial velocity distribution was analysed …; Figure 8-10 show the …; The relative static pressure values of the three convergent vaneless diffusers is formed a stable law after.. ; .. relative position that less than or equal to 60% ..;  ;etc)  
